# Impact of Indigestible Materials on the Efficiency of Fecal Corticosterone Immunoassay Testing in *Pituophis* Species

**DOI:** 10.3390/ani12111410

**Published:** 2022-05-30

**Authors:** Holly Racine, Kinsey Skalican Guthrie, Tyler Hill, Zachary Loughman

**Affiliations:** 1Department of Biomedical Sciences, West Liberty University, West Liberty, WV 26074, USA; holly.racine@westliberty.edu (H.R.); tphill@westliberty.edu (T.H.); 2Department of Organismal Biology, Ecology, and Zoo Science, West Liberty University, West Liberty, WV 26074, USA; kskalican@westliberty.edu

**Keywords:** stress, feces, corticosterone, extraction, enzyme immunoassay, *Pituophis*

## Abstract

**Simple Summary:**

Evaluating stress in animals is important for improving animal welfare and husbandry. However, it has been challenging to establish reliable and noninvasive methods for quantifying stress. Steroid hormones released during prolonged periods of stress are metabolized by the liver and excreted in feces. In snakes, corticosterone is the primary hormone of stress and is often measured in fecal samples collected from these animals. Assays can be used to measure the metabolites of corticosterone from feces, but are limited by compounding factors (season, reproductive status, diet, etc.) that influence the ability to confidently conclude the results. Our overall objective is to standardize these methods by first improving the extraction methodology. We found that 75% of the fecal sample contained indigestible materials (hair, teeth, bone, etc.) and, therefore, interferes with the extraction process. After removing the indigestible material, we found that we had a 95% improvement in overall yield. These findings alleviate one limitation to using fecal samples to measure stress in animals.

**Abstract:**

Measuring fecal glucocorticoid metabolites (FGM) has recently become a sought-after method for assessing stress in animals. While there are many benefits to this methodology, there are also recognized limitations, including the apprehensive interpretation of results. While many factors can influence FGM levels, we aimed to standardize and improve these methods in snakes. Fecal samples were collected from *Pituophis* species and FGMs were extracted by two different sample collection methods: (1) fecal sample containing undigested materials and (2) fecal samples with undigested materials removed. These extracts were then used to quantify FGM concentrations using a corticosterone EIA kit. The results indicated that the samples with the undigestible materials removed had a 95% increase in overall yield (*p* < 0.01). Since the collected fecal samples contain 75% undigestible materials by weight, these results support our hypothesis that removing these materials will improve extraction methods for a more reliable measurement of corticosterone. This is the first step towards standardizing the methods for assessing stress by measuring fecal glucocorticoid metabolites in snakes.

## 1. Introduction

Evaluating stress in an animal is challenging, and the assessment of an animal’s response to a stressor, such as an environmental condition or human-induced disturbances, is often objective. Stress causes disruption in an animal’s physiological homeostasis, and prolonged stress can lead to lasting and overall poor welfare in any animal [1]. Therefore, animal endocrinologists have continued to explore methods for assessing stress in animals both in the wild and in captivity to improve animal welfare and husbandry. More often, stress in mammals, birds, and fish is studied due to the more charismatic nature or agricultural importance of these animals [2]. Understanding and identifying when animals are stressed are important elements of effective welfare practices. In ectotherms, where the internal state is difficult to assess behaviorally, determining when an animal is experiencing acute or chronic stress can be a challenge. 

Upon a stressful stimulus, the initial physiological response in animals includes the arousal of the sympathetic nervous system, which triggers the secretion of catecholamines, including epinephrine (also known as adrenaline), from the adrenal glands into the blood [3]. The secondary response includes the hypothalamus–pituitary–adrenal (HPA) axis, which maintains the release of epinephrine, while also causing the secretion of glucocorticoids (cortisol and corticosterone) into the blood [3,4]. The main glucocorticoid in reptiles is corticosterone and levels can be detected in the blood within minutes [2]. 

The short-term (minutes to days) corticosterone concentration has been measured in blood and saliva [2,5,6]. These methods vary depending on the animal species and available resources. Feces and keratin-rich shed skin can be used to assess chronic (weeks to months) stress [5,7,8,9]. The assessment of chronic stress provides insights into the overall stress response, and is an ideal way to evaluate wild and captive snakes that are confronted with stressors that may impact their overall health. 

One promising method that has become increasingly more favorable to both wildlife biologists and zoologists involves immune-based hormone assays using collected fecal samples to quantify long-term corticosterone levels. Fecal hormone analysis has many benefits compared to other methods, including the following: (1) ease of collection and storage, (2) methods for collection are entirely noninvasive, (3) sampling is attainable for even smaller species, (4) collection of samples does not require training, and (5) reliable assays are commercially available for quantification. However, there are recognized limitations for fecal analyses as well. Since glucocorticoids are metabolized by the liver before excretion, the fecal glucocorticoid metabolite (FGM) levels are measured, instead of the circulating hormones [10]. To avoid discrepancies, reputable assays that are commercially available are designed specifically to measure FGM levels in a variety of excrements, including feces [11,12,13,14]. FGM levels reflect circulating levels of corticosterone [10,15,16]. Other limitations include the difficulty in interpretating the results, since the stressor can be influenced by many factors, including seasonal changes, sex, body condition, diet, reproductive status, and gut passage times, among others [8,17]. Therefore, it is a common goal to standardize and improve these methods.

To avoid confounding factors that may impact the reliability of FGM assays when quantifying levels of corticosterone in fecal samples of snakes, the objective should be to create a database of baseline FGM levels for individual collections of snake species that accounts for factors by normalizing levels based on standard conditions, compared to introduced stressors, such as habitat change, handling, altered feeding, etc. However, before building such a database, the methodology of extracting corticosterone from fecal samples must first be improved to enhance the overall efficiency. 

When measuring corticosterone levels, another common concern is that published results are not reliable, due to the discrepancies caused by the variety of indigestible items in the collected fecal sample [9,14,15,18]. Indigestible materials that are excreted in feces include hair, teeth, and bones. Thus, when measuring samples to quantify fecal concentrations of FGMs, the true mass of excrement may vary depending on the digestive efficiency. Others have recommended removing undigested materials from mammals and birds before FGM extraction protocols to improve reproducibility [14,15]. However, it was our objective to develop methods for purifying the fecal excrement by eliminating indigestible materials to improve the extraction yield in snakes. The digestive efficiency in snakes varies greatly, but may be as high as 99.3–99.8% [17]. However, the indigestible materials that are excreted in the feces are components of the prey item ingested and not the outcome of digestion by the snake. We, therefore, hypothesized that by removing these items from the fecal sample (more hair and teeth, less extracted FGMs per unit mass), 

We would increase both the extraction efficiency and overall yield per unit mass of FGM concentrations. 

## 2. Materials and Methods

### 2.1. Ethics Statement

This study was performed using snake species from the live animal collection of the West Liberty University Zoo Science program. For all animals used, no interventionary studies were performed, and fecal samples were collected by noninvasive approaches. Therefore, no IACUC was required.

### 2.2. Fecal Sample Collection

Eleven fecal samples were used from three *Pituophis* species: *Pituophis melanoleucus lodingi* (black pine snake), *Pituophis ruthveni* (Louisiana pine snake), and *Pituophis catenifer sayi* (bull snake). They were individually housed in vision V70 rack tubs (34.5′′ × 21′′) under LED lighting on a 12 h light cycle in average humidity of 50–60%, with humidity hides available during shedding (Table 1). All of these animals were hatched in 2019 and followed the same diet (one small rat two times a week) and husbandry routine (21 °C nighttime temperature and 31 °C daytime temperature with 2 h ramping between nighttime and daytime temperature changes). Enclosures were checked daily to obtain the freshest possible fecal samples. Animals were not disturbed or handled during collection. Once collected, samples were weighed, labeled, and frozen (−80 °C) until ready for processing. Refer to Table 1 for description of *Pituophis* species used for sample collection. 

### 2.3. Fecal Extraction Methods

Prior to quantifying FGM concentrations, these steroid metabolites first needed to be extracted from our collected fecal samples. Methods described below are adapted from the recommended protocol provided by DetectX^®^ Steroid Solid Extraction Protocol (Arbor Assays^®^, Eisenhower Place, Ann Arbor, MI, USA, K014). Prior to the start of extraction, samples were passively dried at room temperature (20–22 °C) on a reverse fume hood over a period of two days. From each individual, the total dried fecal sample was weighed prior to transferring 0.1 g of the sample with undigested materials (FS+) into a 2 mL vial. The FS+ portion, while randomly selected, was the closest representation of the fecal sample composite. The remaining sample was then crumbled and sifted by hand over a 600 μm test sieve, allowing the dry powdered fecal sample to fall below the mesh and retaining all indigestible material greater than 0.6 mm (Figure 1). The sample, now with sizeable undigested materials removed (FS−), was then weighed for comparison to total FS+ recorded mass, and then 0.1 g of sample was collected into an additional 2 mL vial. This procedure was repeated for each individual. Samples were frozen (−80 °C) if the extraction process was not performed on the same day. All sample extractions occurred within a month of collection.

Fecal glucocorticoid metabolite (FGM) extraction was performed by adding 1 mL of 80% ethanol (1 mL of ETOH/0.1 g of sample). Samples were shaken vigorously in a pulsating vortex mixer (Fisher Scientific) for 30 min. Next, they were placed in a centrifuge (Beckman Coulter Allegra 2IR Centrifuge) for 15 min at 5000 rpm at 4 °C. The supernatant containing the extracted FGM was then carefully removed, being sure not to disrupt the pellet at the bottom, and placed into a 1.5 mL vial. Supernatant volumes were recorded for later analysis of assay tests, and then passively dried overnight. Drying occurred in a gravity convection laboratory oven at 40 °C. Dried supernatant was then resuspended in 100 μL of 80% ethanol. 

To measure extraction efficiency, select fecal samples (N = 6) were divided into 3 additional 0.1 g increments prior to extraction procedures. Alongside an unaltered sample, the 3 additional partitions were spiked with 125 μL of known concentrations (75, 150, and 300 pg/mL) of analyte (corticosterone standard provided by Arbor Assays DetectX^®^ Corticosterone EIA kit). Samples were then subjected to the extraction methods as described above. Extraction efficiency was calculated using the following equation: (amount observed)/amount expected) × 100%, where the amount observed is the value from the measured spike concentration minus the measured unspiked concentration, and the amount expected is the concentration of the spike that was added. Extraction efficiencies were averaged and then compared between FS+ and FS− samples. 

### 2.4. Corticosterone EIA for Quantification of FGM Concentrations 

To quantify FGM concentrations, extracts were prepared for analysis using the Arbor Assays DetectX^®^, Ann Arbro MI, USA Corticosterone EIA Kit. All reagents were prepared following the kit protocols. Samples were diluted (20×) in the prepared assay buffer to ensure ethanol concentration was <5% as instructed. Following the included assay protocol, a colorimetric 96-well microplate reader (Synergy H1 microplate reader) capable of reading optical density at 450 nm was used to measure corticosterone concentrations relative to the standard curve. All samples were run in duplicate, with an intra-assay coefficient of variation of 6.54 and an inter-assay coefficient of variation of 4.37. Final concentrations were calculated from generated optical density readings using the online tool from MyAssays^®^. Dilution factors were accounted for during calculations. Limit of detection for the assay was determined as 16.9 pg/mL.

### 2.5. Data Analysis 

All statistical analyses for this study were performed using R software (R Core Team, version 4.1.3) with a *p*-value below 0.05 as a criterion for significance [19]. A total of 11 individuals were used in this study (d = 1.34, α = 0.05, power = 0.98, *n* = 11). The distribution of each independent variable was assessed by visual inspection of histograms and Q-Q plots in combination with a Shapiro–Wilk test. Residuals were tested for homogeneity of variances (F = 1.5426, *p* = 0.5054). Data were normally distributed and met the assumptions of homoscedasticity. For the comparison of corticosterone concentrations for each individual measured between the FS+ and FS− methods for extraction, the difference in groups was assessed using a paired *t*-test. Additionally, extraction efficiency of the FGM concentrations was measured between the FS+ and FS− methods (*n* = 6) for extraction, and significance was determined using a paired *t*-test. 

## 3. Results

### 3.1. Measuring Proportions of Undigested Materials 

Following filtering of undigested materials, including hair, teeth, and bones, from each fecal sample, the overall intial dry mass of the samples with undigested materials (FS+) versus the measured masses of fecal samples following the removal of undigested materials (FS−) demonstrated that 75% of the total fecal sample (FS+) consists of undigested materials (Table 2). Variation in digestion between individuals results in different sized excrements. Removal of the undigested materials led to a considerable decrease in the overall mass available for analysis. However, regardless of size, the FS− samples were all equally a fine powder in consistency (Figure 1). 

### 3.2. Comparison of Extraction Efficiency between FS+ and FS− Samples 

Extraction efficiency analyses are performed to validate extraction methods, and are often used to calculate corticosterone concentrations more accurately. The extraction efficiency in fecal samples (N = 6) with undigested materials (FS+) demonstrated an average extraction efficiency of 59.2 ± 16.2% (mean extraction efficiency ± SD), while the average extraction efficiency in fecal samples (N = 6) without undigested materials (FS−) demonstrated an average extraction efficiency of 59.2.0 ± 17.3% (mean extraction efficiency ± SD). 

There was no statistically significant difference in extraction efficiency between the methods (t(5)= 0.003, *p* = 0.997). 

### 3.3. Removal of Undigested Materials Leads to an Increased Yield of Corticosterone Concentrations 

Enzyme immunoassays were performed to measure the concentrations of FGMs in both fecal samples with undigested materials (FS+) and fecal samples following the removal of undigested materials (FS−) from the same individual, demonstrating an increase in corticosterone yield with undigested materials removed. While individual samples (N = 11) varied in concentration of corticosterone (Figure 2A), the overall average calculated resulted in a 95% increase in yield in fecal samples following the removal of undigested materials (Figure 2B) (t(10) = 4.445, *p* < 0.01). These findings support our hypothesis that the removal of undigested materials increases the overall yield of corticosterone concentrations detected using enzyme immunoassays. The sample variation can be explained by the limitations in normalizing conditions for each individual animal.

## 4. Discussion

The purpose of this study is to establish methods for extracting fecal glucocorticoid metabolites (FGMs) from fecal samples, specifically from corticosterone metabolism, as a noninvasive alternative to study chronic stress in snakes. The results from this study support our methodology for extraction of FGMs from fecal samples. In regard to extraction efficiency, there was no supported benefit when removing undigested materials from samples prior to extraction, since there was no significant difference between the extraction efficiency of the FS+ and FS− samples (t(5) = 0.003, *p* = 0.9974). This lack of difference is most likely because the extraction efficiency is normalized to the unspiked control within the individual sample group, and the undigested materials may not interfere with our extraction procedures. However, this practice does increase the overall yield in detected levels. Hair and other undigested materials may displace or falsify the useable portion of the fecal sample and could skew the results following corticosterone immunoassay analyses. Removal of the undigested materials negates any argument that the reported levels are skewed by extracted FGM levels from prey items in the fecal sample. This methodology can, therefore, be useful for optimizing the overall yield of FGMs following extraction procedures. This increase in yield will better represent the levels of FGM in collected fecal samples. The optimization of these methods will allow us to proceed with the long-term goal of standardizing individual groups of snakes, accounting for factors including seasonal changes, sex, body condition, diet, reproductive status, and gut passage times, among others. Variation in results occurs when these factors are not accounted for, as observed in our data (Figure 2A). Future studies will account for these factors in order to create a database that will provide baseline levels of FGMs from fecal samples measured in multiple snake taxa under controlled conditions, so that these can be made available when studying the impact of stressors in snakes. 

While the methods for using immunoassay-based analyses in fecal samples are being utilized more and more frequently [5,9,10,11,13,20], there has been some pushback on the reliability of these methods. Concerns with these methods are valid and based on inconsistent reports of hormone levels in fecal samples, caused by storage methods or other physiological influences [6,21,22,23]. However, there are a considerable number of benefits to this methodology, including a noninvasive method of sample collection, as well as the means to assess net effects of stress accumulated over time [8,9,24,25]. This, therefore, supports our objective to validate techniques for hormone analyses and determine these baseline levels. 

Stressors for snakes may be human-induced, but also include environmental factors, such as seasonality and climatological changes, reproductive status, disease, and availability of food and shelter [20,26,27,28]. For example, seasonal effects of hormone levels were observed in a population of pigmy rattlesnakes (*Sistrurus miliarius*). Lind et al. (2018) reported increased levels of corticosterone during the fall months leading up to the winter [28]. This increase in corticosterone during the fall was acknowledged by groups studying other taxa of snakes as well, including red-sided garter snakes (*Thamnophis sirtalis parietalis*) [29]. However, there have been conflicting patterns in seasonality changes reported in studies on other *Thamnophis* species [30]. 

The impact of female reproductive status on corticosterone levels has also been studied. The levels are described to vary depending on their annual cycles (decreased during vitellogenesis and increased during ovulation), in both captive and wild animals [29]. These levels appeared to be the most consistent for wild snakes during the spring months. The variation in levels reported from captive animals may also be influenced by other factors, such as human-induced stressors. A study by Neuman-Lee et al. determined that even acute levels of stress (such as sample collection) can increase corticosterone levels [30]. This same group also reported a significant increase in baseline corticosterone levels measured in males compared to females [30]. Therefore, both sex and seasonality can affect stress levels. Our study aimed to compare methods for measuring FGM levels and, therefore, we did not account for sex and seasonality. However, we will ensure that further use of these methods will account for the effects of these factors.

This substantial variability in corticosterone detection also explains our results shown in Figure 2. While all individuals were approximately the same age and *Pituophis* species, and samples were collected in the late summer/early fall, the limitations in our collection meant that there was no control for sex, mass, or reproductive status. Since our study focused on comparisons between extraction methods, each individual served as its own control and provided a pair of samples for analysis. In future studies directed towards establishing a baseline for FGM levels, our sample groups will account for possible physiological variation in order to validate methods.

## 5. Conclusions

Our study revealed a significant increase in overall yield in the concentration of fecal glucocorticoid metabolites (FGM) measured in fecal samples collected from *Pituophis* species following the elimination of indigestible materials, including hair, teeth, and bone, from the sample prior to extraction. While there are still many factors to consider before standardizing our overall methodology, we now have an established method for FGM extraction. Future studies will continue to support the benefits of using fecal samples to assess stress. 

## Figures and Tables

**Figure 1 animals-12-01410-f001:**
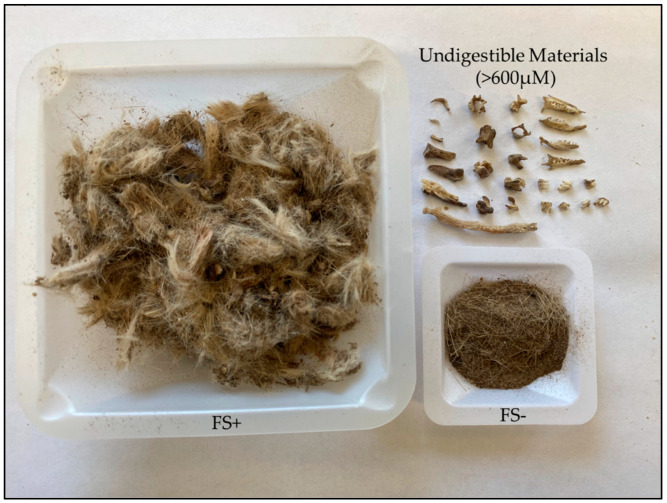
Captured image of separated fecal remains with undigestible materials removed. Dried sifted fecal sample (lower right corner; FS−) was reduced compared to the fecal sample containing undigestible materials (left; FS+). Undigestible materials consist of keratinized materials including hair, toenails and bone; >600 μM (upper right corner).

**Figure 2 animals-12-01410-f002:**
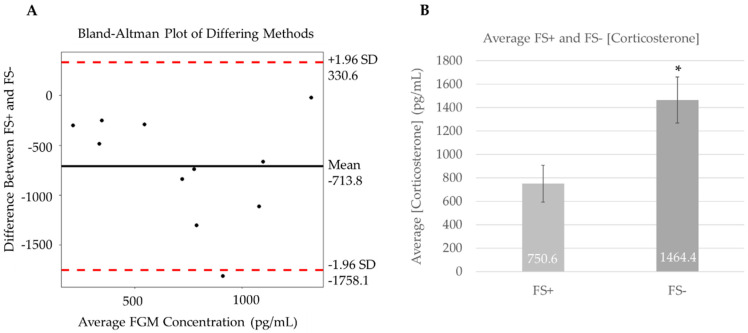
Difference in extraction methods for measuring corticosterone from fecal samples of *Pituophis* species. (**A**) Bland–Altman plot displays the difference in measured corticosterone (N = 11) comparing yield from different fecal sample extraction methods: one from undigested materials (FS+) and the other with undigested materials removed (FS−). (**B**) Bar graph (error bars represent SEM) emphasizing the 95% increase in overall yield of corticosterone measured from FS− compared to FS+ (* *p* < 0.01).

**Table 1 animals-12-01410-t001:** Description of *Pituophis* species used for sample collection during study.

Individual	*Pituophis* Species	Sex	Age (Years)
1	*catenifer savi*	Female	2
2	*catenifer savi*	Male	2
3	*melanoleucus*	Female	2
4	*ruthveni*	Male	2
5	*catenifer savi*	Male	2
6	*catenifer savi*	Male	2
7	*melanoleucus*	Female	2
8	*catenifer savi*	Male	2
9	*ruthveni*	Male	2
10	*catenifer savi*	Male	2
11	*catenifer savi*	Male	2

**Table 2 animals-12-01410-t002:** Masses of collected fecal samples. Masses from individual fecal samples prior to removal of undigested materials (FS+) and following their removal (FS−).

Sample	FS+ (g)	FS− (g)	Undigested Materials (%)
1	3.06	0.89	70.9
2	5.92	2.3	61.1
3	4.85	1.5	69.1
4	4.21	1.13	73.2
5	3.24	0.61	81.2
6	3.68	0.98	73.4
7	5.1	1.63	68.0
8	3.09	0.94	69.6
9	5.18	0.83	84.0
10	5.84	0.89	84.8
11	7.13	0.82	88.5

## Data Availability

Data is available upon request from the primary author of this work.

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
