# Peer review of "Impact of Indigestible Materials on the Efficiency of Fecal Corticosterone Immunoassay Testing in Pituophis Species"

_animals, 2022, doi:10.3390/ani12111410_

Round 1

Reviewer 1 Report

The authors present the results of an investigation into the effects of faecal sample preparation on FGM yield from captive snakes. This work has important practical implications and could help to hone tools needed to beter understand and improve the welfare of captive snakes, as well as understanding of the biology of free living animals. 

In its current state, however, the MS requires substantial improvement in order to be accepted for publication, in my view. Even should the concerns listed below, and on the attached MS, be addressed, I believe that the work may be better suited for another jurnal as, given its sample size, it represents more of a pilot study than a full blown research study. For exmaple, the 'forgotten species' special issue of the mdpi journal JZBG could be a good venue for the work.

Key concerns/issues (see MS comments for more details and for additonal comments):

  1. The methods are not clear; it is very difficult to follow the steps of what was done, and little literature is cited to underpin the chosen methods. Please see comments on the MS for details of where I believe more clarity is needed. Perhpas it's because I am not an endocrinologist, but the extraction efficiency methods are particularly unclear to me and I really don't understand what was done in this partof the study, based on the described methods. Most usefully, the authors could include an opening Methods paragraph explaining the overal approach, and defining key datasets produced.
  2. Statistical analyses are not fully described or justified. Degrees of freedom are absent from reported stats outcomes, some tests results are reported without any details provided in the methods. Assumptions of tests (normality, homogeneity of variance) are not addressed. The undescribed test comparing males and females, mentioned in the Discussion, almost certainly does not have a large enough sample size to be valid. 
  3. The authors do not discuss the implications of the yield differences they detected for interpreting this sort of data in snakes. They also do not dicsuss the enormous vairation in the differences between sample types across individuals; this ranges from near-zero to almost an order of magnitude. Bland-Altman and associated analyses to investigate the agreement between methods wouldbe really useful here to partition the variation into random and systematic biases. This would then allow the authors to talk about whether results produced from unfiltered samples could be compared with those from filtered samples with any confidence. 
  4. The authors do not discuss the validation of these techniques through comparison with blood hormone levels and/or in response to known stressors. Based on their data, we know that the filtering of samples makes a potentially large difference in results, but we don't know which one is actually more representaitve of circulting hormone levels. It makes sense that it would be the filtered results, but this is assumption rather than proven fact.
  5. Include your raw data as additional columns in  Table 2

Reviewer 2 Report

Review of the article: Impact of Indigestible Materials on the Efficiency of Fecal Corticosterone Immunoassay Testing in Pituophis Species

Manuscript ID: animals-1680771

In this study, the authors aimed to standardize and improve the methods for measuring fecal glucocorticoid metabolites (FGM) in snakes and tested the hypothesis that removing the undigested materials will improve extraction methods for a more reliable measurement of corticosterone. The authors need to notice and properly reply the following comments.

Major Compulsory Revisions:

  1. The undigested materials (e.g., hair and toe nails) in the fecal samples need to be removed (Bienboire-Frosini et al. 2018; Palme et al. 2013) before FGM extracting procedures because these materials are part of the prey’s body and not digested or metabolized by the snake. Such materials must not be added in the samples for the FGM measurements. That is, comparing the FGM extracting by two different sample collection methods (i.e., with or without undigested materials) is not the most appropriate or important topic of methodological studies, because only fecal samples with undigested materials removed will be acceptable samples for FGM measurements. Comparing the FGM extracting by the two methods may be one subtopic of a study but not the main topic.

References:

Bienboire-Frosini, C.; Alnot-Perronin, M.; Chabaud, C.; Asproni, P.; Lafont-Lecuelle, C.; Cozzi, A.; Pageat, P. 2018. Assessment of Commercially Available Immunoassays to Measure Glucocorticoid Metabolites in African Grey Parrot (Psittacus Erithacus) Droppings: A Ready Tool for Non-Invasive Monitoring of Stress. Animals 8(7): 105. https://doi.org/10.3390/ani8070105

Palme, R.; Touma, C.; Arias, N.; Dominchin, M.F.; Lepschy, M. 2013. Steroid extraction: Get the best out of faecal samples. Wiener Tierärztliche Monatsschrift. Vet. Med. Austria 100: 238–246.

  1. Detailed descriptions of the animals and housing should be presented in the Materials and Methods, including cage size, food type, feeding frequency/amount, environmental factors (temperature/humidity/illumination/shelter/seasonality), disease, reproductive status, and husbandry procedure, as this study was not reviewed or permitted by IACUC.
  2. At lines 132-133, the authors mentioned that three additional samples were spiked with known concentrations (75, 150, 300 pg/mL) of corticosterone extract. How were the four EIA results originally from the same feces used in the calculation of extraction efficiency by the equation listed at lines 142-143? Why using an unpaired t-test here? What were the results of the normality test and homoscedasticity test of data? What was the sample size (N=6; at line 173 or 174) defined here? In addition, what were the intra- and interassay coefficients of variation of samples and what was the detection limit of the FCM analysis?

Minor Revisions:

  1. At line 3: “species” should be “Species”.
  2. At line 40: “animals” should be “animal”.
  3. At line 75: “FGM levels reflect circulating levels of corticosterone” should be supported by citing some literatures.
  4. At lines 111, 113: “classification” should be revised because not merely scientific names are shown in Table 1.
  5. At line 113: “collected” should be “collection”?
  6. At lines 116-117: How to randomly select the location on each feces for sampling?
  7. At line 119: “retain” should be “retaining”?
  8. At line 163: “Collected Fecal Samples” should be “collected fecal samples”.
  9. At line 169: “corticosterome immunassay” should be “corticosterone immunoassay”.
  10. At line 175: “digested” should be “undigested”.
  11. At line 183: “and” should be “an”?
  12. At line 193: “INDIVUAL” should be “INDIVIDUAL”.
  13. At line 197: “the FS- samples increased the concentration” should be revised.
  14. At line 200: Why not presenting the variations (by showing error bars) of each group in Figure 3?
  15. At line 204: The data for “high standard deviation” were not shown here.
  16. At line 207: “extraction” should be “extracting”.
  17. At line 239: “LA” should be removed.
  18. At lines 314-315: Some information of this reference is missed.

Round 2

Reviewer 1 Report

Thanks to the authors for addressing the points I raised regarding their manuscript. I am satisfied with their responses, other than in one place.

The authors now report that data distribution was visually inspected via histograms and QQ plots, and Shapiro-WIlkes test for normality was performed, but don't actually report what was found. Please could they report the results of the SW test, and confirm that the vidual inspection of the data was consistent with normal distribution. 

The authros also have not tested the other main assumption of T tests, that is homoscedasticity. Please could the authors perform a Levene's test or equivalent to compare the variance in each sample?

Provided that the above points are addressed, I am supportive of publication.

Reviewer 2 Report

Revised version review of the article: Impact of Indigestible Materials on the Efficiency of Fecal Corticosterone Immunoassay Testing in Pituophis Species

Manuscript ID: animals-1680771

The authors have addressed some of my comments. The following points should be also addressed before publication.

  1. I suggest the authors to mention in the Introduction or Discussion that the undigested materials in the fecal samples need to be removed before FGM extracting procedures because these materials are part of the prey’s body and not digested or metabolized by the snake. Such materials must not be added in the samples for the FGM measurements.
  2. Some detailed descriptions of the animals and housing are still not presented in the Materials and Methods, such as cage size, humidity, and illumination, as this study was not reviewed or permitted by IACUC.
  3. What were the results of the normality test and homoscedasticity test of data for t-tests?
  4. What were the intra- and interassay coefficients of variation and what was the detection limit of the FCM analysis in this study?
  5. At line 125: How to “randomly” or “evenly” select and transfer 0.1g of the fecal sample with undigested materials as they are not homogeneous?
  6. At line 158: “includinghair andtoe”?
  7. At line 202: “59.2.0% (±17.3)”?
  8. Figure 2B: What do the error bars means (e.g., SD or SE)?
  9. Some information in the references 14, 18, and 24 are missed.
